# Approximation Model Development and Dynamic Characteristic Analysis Based on Spindle Position of Machining Center

**DOI:** 10.3390/ma15207158

**Published:** 2022-10-14

**Authors:** Ji-Wook Kim, Dong-Yul Kim, Hong-In Won, Yoo-Jeong Noh, Dae-Cheol Ko, Jin-Seok Jang

**Affiliations:** 1Dae-Gyeong Division, Korea Institute of Industrial Technology, Daegu 42994, Korea; 2Gyeongbuk Research Institute of Vehicle Embedded Technology, Yeongcheon-si 38822, Korea; 3Department of Mechanical Engineering, Pusan National University, Pusan 46241, Korea; 4Department of Nanomechatronics Engineering, Pusan National University, Pusan 46241, Korea

**Keywords:** machining center, vibration test, dynamic stiffness, approximate model, design of experiments

## Abstract

To evaluate the dynamic characteristics at all positions of the main spindle of a machine tool, an experimental point was selected using a full factorial design, and a vibration test was conducted. Based on the measurement position, the resonant frequency was distributed from approximately 236 to 242 Hz. The approximation model was evaluated based on its resonant frequencies and dynamic stiffness using regression and interpolation methods. The accuracy of the resonant frequency demonstrated by the kriging method was approximately 89%, whereas the highest accuracy of the dynamic stiffness demonstrated by the polynomial regression method was 81%. To further verify the approximation model, its dynamic characteristics were measured and verified at additional experimental points. The maximum errors yielded by the model, in terms of the resonant frequency and dynamic stiffness, were 1.6% and 7.1%, respectively.

## 1. Introduction

Recently, the demand for self-optimization technology to be used in manufacturing systems for optimizing process conditions, predicting failure, managing prognostics, and improving product quality has increased. Manufacturing system intelligence technology applied with self-optimization technology enables the development of production processes for various product models and the derivation of stable optimal processing conditions. An intelligent processing system virtualizes the machining process by combining predictive models with real-time data. Heo et al. combined the measured data with normal and abnormal monitoring systems using the pre-predicted cutting shape, calculated using the virtual machining software as an anomaly detection criterion [1]. It is a convergence technology that predicts the machining process in real time via data learning and determines the optimal machining conditions, based on the equipment, material, process, and tool characteristics. Choi et al. extracted the features that are most sensitive to tool wear by the signal analysis measured using multiple sensors. Through this, they developed and evaluated the architecture of a real-time tool wear monitoring system using a multilayer perceptron neural network. Kim et al. [2] performed energy monitoring of each axis by comparing the sum of the DC power consumed by each axis with the sum of the PMC AC power supplied to each inverter. Through this, they developed an energy monitoring system for each axis of the machine tool to determine whether chatter, wear, and abnormal processes occur. Lee et al. [3] developed a surface shape prediction system that considers cutting factors for the development of a virtual turning system. In addition, a Watchdog Agent system was developed to evaluates and predict the performance and status of the system in real time during the processing process, and the feature extraction method representing the system performance from the measured signal was studied. Lee [4] suggested the angular contact bell bearing as being the main cause of failure of a machine tool spindle and analyzed the failure mechanism. They obtained the vibration data of the spindle through an accelerated life test and developed a condition diagnosis algorithm, using it to predict the life of the spindle. Kim et al. [5] analyzed normal and abnormal signals by measuring acceleration data generated during cutting. This paper proposed a fault diagnosis algorithm that applied a CNN algorithm using a raw signal containing external noise, without using FFT or other filters [6]. The intelligent processing system to which self-optimization technology is applied comprises virtual processing, a learning model, a monitoring system, and machinability diagnostic control system technology. The virtual machining system predicts the cutting load in real time via the development of a machining process simulator for field workers, as well as calculates and provides the contact stiffness of tools and materials.

To develop a virtual machining prediction model and improve accuracy, the dynamic characteristics of the machining equipment must be analyzed. Altintas evaluated the cutting stability using the transfer function between the tool and workpiece at the tangent and normal lines in milling. For stability evaluation, the contact coefficient between the machining tool and workpiece was considered. According to Rehorn et al. [7], the quality of high-precision parts largely depends on the performance of the machining system, and it is determined by the mechanical structure of the machine tool and the interrelated dynamics of the cutting process. The dynamics of the combined spindle/cutter system, which is a major component of any machine tool, was identified using a modal test. This study concluded that for precision workpieces, the dynamics of the spindle and cutter system must be considered to improve future machining controls and processes. Ozsahin et al. [8] used the frequency response function of the tool point for chatter analysis. The centrifugal force, due to the gyroscope moment generated during the cutting operation, and the bearing dynamics thereby change the FRF. Bearing dynamics and mathematical models based on working conditions are also presented. The tool point FRF, determined computationally using the parameters of the bearing, was verified by chatter testing [9]. 

The dynamic characteristics of a machining equipment changes, depending on the tool used. In addition, as the main spindle is transferred for cutting, the mass distribution of the equipment changes, and the dynamic characteristics change accordingly. Because the dynamic characteristics of the machining equipment, based on the position of the main spindle, are not constant and exhibit nonlinearities, the dynamic characteristics at all positions of the main spindle must be analyzed. However, measuring the dynamic characteristics of all positions of the main spindle and applying them to the prediction model is difficult because of the significant number of experiments required. In this study, the design of experiments is performed to analyze the dynamic characteristics, based on the position of the main spindle of a machining equipment, and the experimental position is selected. Vibration tests are performed at the selected experimental points, and the dynamic characteristics are analyzed. Subsequently, an approximation model is developed, and the approximation model is verified based on additional experimental points.

## 2. Dynamic Characteristics of Machining Center

A vibration test was performed to confirm the change in dynamic characteristics, based on the position of the main spindle of a machining equipment. According to Lee et al., the stiffness of the entire machine affects the high-precision and quality. However, because there is no regulation on the dynamic characteristics, a method for evaluating static and dynamic stiffness was developed. The validity was confirmed by comparing the experimental results with the finite element analysis results. The position of the spindle and compliance characteristics, with respect to the X, Y, and Z axis directions, were analyzed. Choi et al. [10] measured the compliance function using a random excitation test method to evaluate the stiffness of a complex multi-function lathe for crankshaft machining, analyzed the static and dynamic stiffness, and compared it with the FEM analysis. The difference between the analysis value and the measured value of stiffness is the error caused by simplifying the coupling of various parts, such as the feed guide surface and ball screw. Kang et al. [11] analyzed and compared the structural stiffness and dynamics of machine tools using the impact test and excitation test methods. In addition, the accuracy of the natural frequency analysis and the static and dynamic stiffness evaluation methods were verified. It is possible to more accurately predict compliance using the excitation test method, compared with the impact test method. In the case of the impact test method, it is difficult to accurately predict the stiffness because the compliance is evaluated using a linear analogy method [12]. An evaluation system for the static and dynamic stiffness of the machine tools was established, and an imitation tool, a jig, was developed. Using an exciter, we excited the frequencies of several bands, without considering the various unpredictable boundary values, and performed an objective evaluation. The reproducibility of the stiffness evaluation method was evaluated by comparing the compliance using the exciter with finite element analysis. Kim et al. [13] studied the rotational FRF used in the RCSA to analyze the change in dynamic characteristics, according to the tool of the machining center. A modal peak picking method was used for evaluating the dynamic characteristic of combined receptance using RCSA [14]. The machining equipment used in the experiment is shown in Figure 1, and a three-axis machining center of Doosan was used. In the three-axis machining center, the main spindle was transferred in the X, Y, and Z directions, and the transferable distances were 560, 430, and 570 mm, respectively. An exciter and accelerometers were installed at the tip of the tool, and vibration experiments were performed based on the position of the main spindle, as shown in Figure 2. The specifications of the excitation are listed in Table 1, and the experimental conditions are listed in Table 2. The position of the main spindle was measured after transferring 100 and 200 mm in the X-, Y-, and Z-axis directions from the absolute origin of the machining equipment. The frequency response function (FRF), based on the transfer of the main spindle, is illustrated in Figure 3. The transfer of the three axes indicates that the changes in the resonant frequency and stiffness were insignificant; however, the resonant frequency and stiffness changed, depending on the transfer.

The stiffness of the machining can be classified as static and dynamic stiffness, which can be derived using compliance. To derive compliance, the equation of motion in a general system is given by Equation (1).
(1)$$\left[M\right]\left\{\ddot{u}\right\}+\left[C\right]\left\{\dot{u}\right\}+\left[K\right]\left\{u\right\}=\left\{F^{a}\right\}$$
where [M] is mass matrix of structure, [C] is damping matrix, [K] is stiffness matrix, $\left\{\ddot{u}\right\},\;\left\{\dot{u}\right\},\;and\;\left\{u\right\}$ represent the acceleration and velocity displacement of the node, respectively, and $\left\{F^{a}\right\}$ represents the load. This is expressed in the form of a complex number, as shown in Equation (2).
(2)$$\left(\left[K\right]-\omega^{2}\left[M\right]+i\omega \left[C\right]\right)\left(\left\{u_{1}\right\}+i\left\{u_{2}\right\}\right)=\left\{F_{1}\right\}+i\left\{F_{2}\right\}$$
where $u_{1}=ucos\phi ,\;u_{2}=usin\phi ,\;F_{1}=Fcos\phi ,\;and\;F_{2}=Fcos\phi$, and ϕ represents the phase angle. Stiffness can be obtained using the real and imaginary parts of the displacement. The compliance for each frequency is the same as in Equation (3), and the dynamic stiffness can be obtained as the reciprocal of the compliance. In Equation (3), *c* represents a complex matrix or vector.
(3)$$\left(G_{Dyna}\right)=\left[\frac{1}{K_{c}}\right]=\frac{\left\{u_{c}\right\}}{\left\{F_{c}\right\}}\;$$

## 3. Design of Experiments

The dynamic characteristics changed based on the position of the main spindle, and experiments were conducted on a three-axis machining center to develop an approximate model. To obtain maximum information via the minimum number of experiments, a design of experiments was performed. The design of experiments included full factorial design (FFD), central composite design (CCD), an orthogonal array (OA), and Taguchi methods, and FFD and CCD are recommended when random errors exist, such as in the actual experiments [16,17,18]. The characteristic values for the design of experiments are dynamic stiffness and resonant frequency, and the factors are shown in Figure 4. The design of experiments was performed in the X and Z directions. The experimental points obtained using FFD and CCD are shown in Figure 5.

Correlation analysis was performed to evaluate the linear relationship between the factors and characteristic values. The correlation coefficient was the same as that shown in Equation (4). A correlation coefficient of 0.6 or higher indicates a linear correlation; however, if it is 0.6 or less, a nonlinear relationship cannot be determined. In Equation (4), *cov(X, Y)* represents the covariance of two factors (X and Y), as well as the standard deviation [19]. The correlation analysis, based on the FFD and CCD of the characteristic values and factors, is shown in Figure 6. For the FFD case, the resonant frequency and dynamic stiffness exhibited a linear relationship, with a correlation coefficient of 0.6 or higher, based on the transfer of the main spindle in the Z direction. For the CCD case, the position transfer of the main spindle did not indicate a linear relationship.

Analysis of means (ANOM) was performed to determine the relative importance of the factors, based on the average value for each level of each factor, as shown in Figure 7. In the ANOM, the greater the change in the slope and the standard deviation, the more important the factors indicated [19,20,21]. Based on the transfer of the main spindle in the X and Z directions, the standard deviations of the frequencies were 0.8228 and 1.267, respectively, and the standard deviations for the dynamic stiffnesses were 327.7 and 678.9, respectively. The standard deviation in the Z direction and the slope were large. Through correlation and mean analyses, the factor level was confirmed to be relatively small, and the response indicated a linear trend. The experiment was performed using an experimental design method.
(4)$$\mathrm{Correlation}\;\mathrm{coefficient}\;:\rho_{X,Y}=cov\left(X,\;Y\right)/\rho_{X}\rho_{Y}$$

## 4. Approximation Model

### 4.1. Modal Test on Experimental Point

An approximate model was developed to analyze the resonant frequency and dynamic stiffness at all positions of the main spindle. For the approximate model, vibration experiments were performed on the selected experimental points using the design of experiments. To verify the approximate model, a comparative analysis with the actual model was performed for additional experimental points. The experimental points were measured by transferring 0, 300, 600, and 900 mm in the X direction of the main spindle and 0, 150, 300, and 450 mm in the Z direction, and additional experimental points were measured at (X, Z) locations of (150, 375), (450, 225), (750, 75), and (750, 375) mm. The experimental and additional experimental points are shown in Figure 8. For the vibration experiment, an exciter and accelerometer were used at the end tip of the tool, and the experimental conditions are listed in Table 3.

Based on the measurement position, the first resonant frequency for the experimental point was distributed from 236 to 242 Hz, as shown in Figure 9. As the location of the main spindle increased in the Z direction and decreased in the X direction, the resonant frequency increased. The dynamic stiffness was distributed from 6500 to 9000 N/mm, as shown in Figure 10. The dynamic stiffness decreased in the positive Z direction, whereas the stiffness increased in the negative X direction. Compared with the natural frequency, the dynamic stiffness was inversely proportional to the Z direction and proportional to the X direction.

### 4.2. Apporximate Model 

The approximation models are classified into regression and interpolation models, as shown in Figure 11. The regression model is used when the experimental results contain a certain amount of error, many experimental points are involved, and an approximate function representing the overall trend of the experimental points is generated. A representative model of the regression model is the polynomial regression model, the form of which is determined in advance, and the coefficient values of the selected polynomial are determined using the least-squares method. However, it does not express nonlinearities well. The polynomials used in the global model of polynomial regression are given by Equation (5).


(5)
$$\left\{ \begin{array}{c} \;Linear\;model\;:f(x)=b_{0}+\sum_{i=1}^{n}b_{i}x_{i}\\ Simple\;Quadratic\;model\;:f\left(x\right)=b_{0}+\sum_{i=1}^{n}b_{i}x_{i}+\sum_{i=1}^{n}b_{ii}x_{i}^{2}\\ Full\;Quadratic\;model:f\left(x\right)=b_{0}+\sum_{i=1}^{n}b_{i}x_{i}+\sum_{i=1}^{n}\sum_{j=1}^{n}b_{ij}x_{i}x_{j}\\ Simple\;Cubic\;model\;:f\left(x\right)=b_{0}+\sum_{i=1}^{n}b_{i}x_{i}+\sum_{i=1}^{n}\sum_{j=1}^{n}b_{ij}x_{i}x_{j}+\sum_{i=1}^{n}b_{iii}x_{i}^{3} \end{array}\right.$$


The interpolation model does not contain numerous experimental points and generates a smooth approximation function that passes through each experimental point. The interpolation model is representative of the Kriging model, as shown in Figure 12. Kriging expresses the nonlinearities well; however, these nonlinearities are difficult to generate because the global optimization problem must be solved. The polynomials used in the global Kriging model are given by Equation (5). It is composed of the sum of the global model and localized deviation, as expressed in Equation (6). In Equation (6), y(x) represents the Kriging model, *γ* represents the regression coefficient, and f(x) represents the global model, which is typically expressed as a polynomial function. In addition, z(x) represents the local deviation and reflects a normal distribution, with a mean of zero and variance $\sigma^{2}$ [22,23,24,25,26].
(6)y(x)=γf(x)+z(x) 

The statistical characteristics of *z(x)* are the covariance between random variables corresponding to two different points in the design space [27,28,29,30].
(7)$$COV\left[z\left(x^{i}\right),\;z\left(x^{j}\right)\right]=\sigma^{2}R\left(\theta_{k},x^{i},x^{j}\right)$$
where $R\left(\theta ,\;x^{i},x^{j}\right)$ represents the correlation function between two sample points, and θ is the relevant parameter.

The correlation function models commonly used in the Kriging model include the exponential, normal, linear, spherical, cubic, and spline function models. The correlation function model, with the normal model, can provide a relatively smooth and infinitely differentiable surface; thus, it is widely used in engineering applications as a variation function. The normal correlation function is the same as shown in Equation (8).
(8)$$R\left(\theta_{k},x^{i},x^{j}\right)=exp\left[-\sum_{k=1}^{N}\theta_{k}\left|x_{k}^{i}-x_{k}^{j}\right|\right]$$

As the samples set $\left[X_{N\times n}|X_{N\times 1}\right]$, the unknown response value $\tilde{y}\left(x\right)$ at the point x, can be estimated according to the kriging model.
(9)$$y\left(x\right)=f{\left(x\right)}^{T}\gamma +r{\left(x\right)}^{T}R^{-1}\left(Y-y\gamma \right)$$
where
(10)$$R=\left[\begin{array}{ccc} R\left(x^{1},x^{1}\right) & \cdots & R\left(x^{1},x^{N}\right)\\ \vdots & \ddots & \vdots \\ R\left(x^{N},x^{1}\right) & \cdots & R\left(x^{N},x^{N}\right) \end{array}\right]$$
(11)$$\gamma ={\left[\gamma_{1},\;\gamma_{2},\;\cdots ,\;\gamma_{p}\right]}^{T}$$
(12)$$f\left(x\right)={\left[f_{1}\left(x\right),\;f_{2}\left(x\right),\;\cdots ,\;f_{p}\left(x\right)\right]}^{T}$$
(13)$$y={\left[f{\left(x^{1}\right)}^{T},\;f{\left(x^{2}\right)}^{T},\;\cdots ,\;f{\left(x^{N}\right)}^{T}\right]}^{T}$$
(14)$$r\left(x\right)={\left[R\left(x,x^{1}\right),\;R\left(x,x^{2}\right),\;\cdots ,\;R\left(x,x^{N}\right)\right]}^{T}$$

According to the predicted optimal linear unbiased estimation, the undetermined coefficients γ can be obtained.
(15)$$\gamma ={\left(y^{T}R^{-1}y\right)}^{-1}y^{T}R^{-1}Y$$

According to the maximum likelihood estimation method, $\theta_{k}$ is calculated by solving the unconstrained optimization problem
(16)$$\underset{\theta_{k}>0}{\max}\left[-\frac{1}{2}\left(N\left(ln\sigma^{2}\right)+\ln\left|R\right|\right)\right]$$

The variance $\sigma^{2}$ can be expressed as
(17)$$\sigma^{2}=\frac{1}{N}{\left(Y-y\gamma \right)}^{T}R^{-1}\left(Y-y\gamma \right)$$

### 4.3. Approximate Model Analysis

The accuracy based on the polynomial order of the regression and interpolation models for the resonant frequency is shown in Figure 13. The accuracy of the approximate model was evaluated using the r-squared method, as shown in Equation (18). In the r-square, the closer it is to 1, the higher is the accuracy. In this study, the closer it is to 100%, the higher the accuracy, in terms of percentage.
(18)$$R^{2}=1-\frac{\sum_{i=1}^{n}{\left(y_{i}-\hat{y_{i}}\right)}^{2}}{\sum_{i=1}^{n}{\left(y_{i}-\bar{y}\right)}^{2}}$$
where $y_{i}$ is true value, $\hat{y_{i}}$ is predicted value of the approximate model, and $\bar{y}$ is average value of true response.

For the regression model, regardless of the polynomial order, all models indicated an accuracy of less than 60%. The kriging model demonstrated high accuracy in linear and simple quadratic models; in particular, it exhibited an accuracy of approximately 89.1% in simple quadratic models.

The accuracy, based on the polynomial order of the regression and interpolation models for the dynamic stiffness model, is shown in Figure 14. The accuracy indicated by regression, interpolation, and polynomials for the dynamic stiffness was similar. The regression linear and simple quadratic models showed high accuracy; in particular, the regression linear model demonstrated the highest accuracy of approximately 81.1%.

The kriging simple quadratic model was used to investigate the resonant frequency (see Figure 15), and the linear regression model was used to investigate the dynamic stiffness (see Figure 16). The tendency of the natural frequency approximation model was similar to that of the actual model, although an error was indicated at the end. The actual model created a pole at the end, and the resonant frequency changed in the form of a quadratic function. However, for the approximate model, no data were indicated after the endpoint. Because the kriging global model varied linearly, it failed to estimate data at the endpoints. The dynamic stiffness model exhibited a similar overall trend.

### 4.4. Approximate Model Verification

The accuracy of the approximate model was verified using the additional experimental points measured, in addition to the experimental points measured using the design of experiments. Details regarding the additional experiment are presented in Section 4.1. A comparison between the actual and approximation models for the additional experimental points of resonant frequency and dynamic stiffness is presented in Table 4 and Table 5, respectively. Both the resonant frequency and dynamic stiffness are suitable for the approximate model. The resonant frequency indicated a maximum error of 1.6%, although its error was typically less than 1%. Meanwhile, the dynamic stiffness indicated a maximum error of 7.1%; however, its accuracy was higher than the error of the experimental point of the approximate model. Although the resonant frequency changed rapidly at the endpoint and differed from that of the actual model, the additional experimental points indicated a higher accuracy because they reflected the linearity of the polynomial, since no rapid changes occurred and all points were internal.

## 5. Conclusions

Vibration experiments were performed based on the transfer of the main spindle of machining equipment, and the dynamic characteristics were analyzed. The results showed that the dynamic characteristics changed, depending on the transfer of the main spindle. An experimental point was selected using the design of experiments to develop an approximate model. In the design of experiments, FFD and CCD are recommended when random errors exist. In the correlation analysis, the FFD showed a linear relationship with the resonant frequency and dynamic stiffness (based on the Z direction transfer); however, the CCD did not indicate such a linear relationship. The ANOM confirmed that the resonant frequency and dynamic stiffness imposed greater effects in the Z direction. Therefore, in this study, an experimental method was selected based on a two-factor, four-level FFD in the X and Z directions, and vibration experiments were performed. The resonant frequency and dynamic stiffness were evaluated based on the transfer of the main spindle, and an approximate model was developed, compared, and analyzed using polynomial regression and kriging interpolation. The accuracy of the resonant frequency demonstrated by the kriging simple quadratic model was 89.1%, whereas the accuracy of the dynamic stiffness demonstrated by the linear regression model was 81.1%. Verification was performed on additional experimental points using an approximate model. In terms of the resonant frequencies and dynamic stiffness, errors of approximately 1.6% and less than 7.1% were indicated, respectively. The developed approximate model was validated using additional experimental points.

The simple quadratic Kriging model is suitable for the approximation model of the resonant frequency, and the linear regression is suitable for the approximation model of the dynamic stiffness.The results of this study show that the dynamic characteristics changed according to the position of the main spindle for the two types of machining equipment, although the difference was insignificant.The dynamic characteristics differed significantly under extreme machining conditions, such as when the main spindle was located at the end.The change in the dynamic characteristics of the system is insignificant in the main work area.Fine dynamic characteristic changes should be considered for high-precision processing; however, the changes in the dynamic characteristics during processing, in most main work areas, are negligible.

## Figures and Tables

**Figure 1 materials-15-07158-f001:**
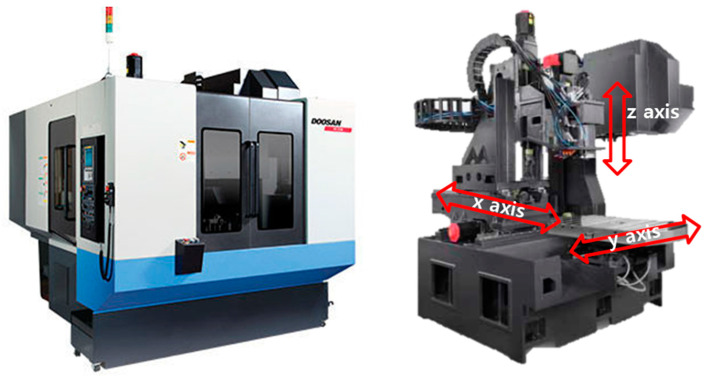
Three-axis machining center.

**Figure 2 materials-15-07158-f002:**
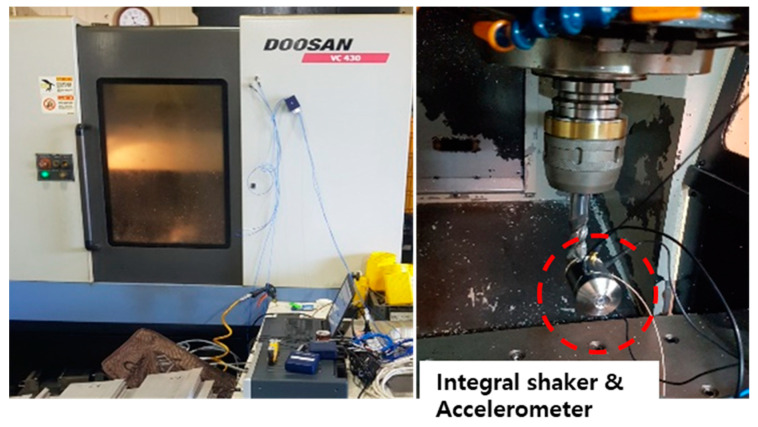
Experimental setup.

**Figure 3 materials-15-07158-f003:**
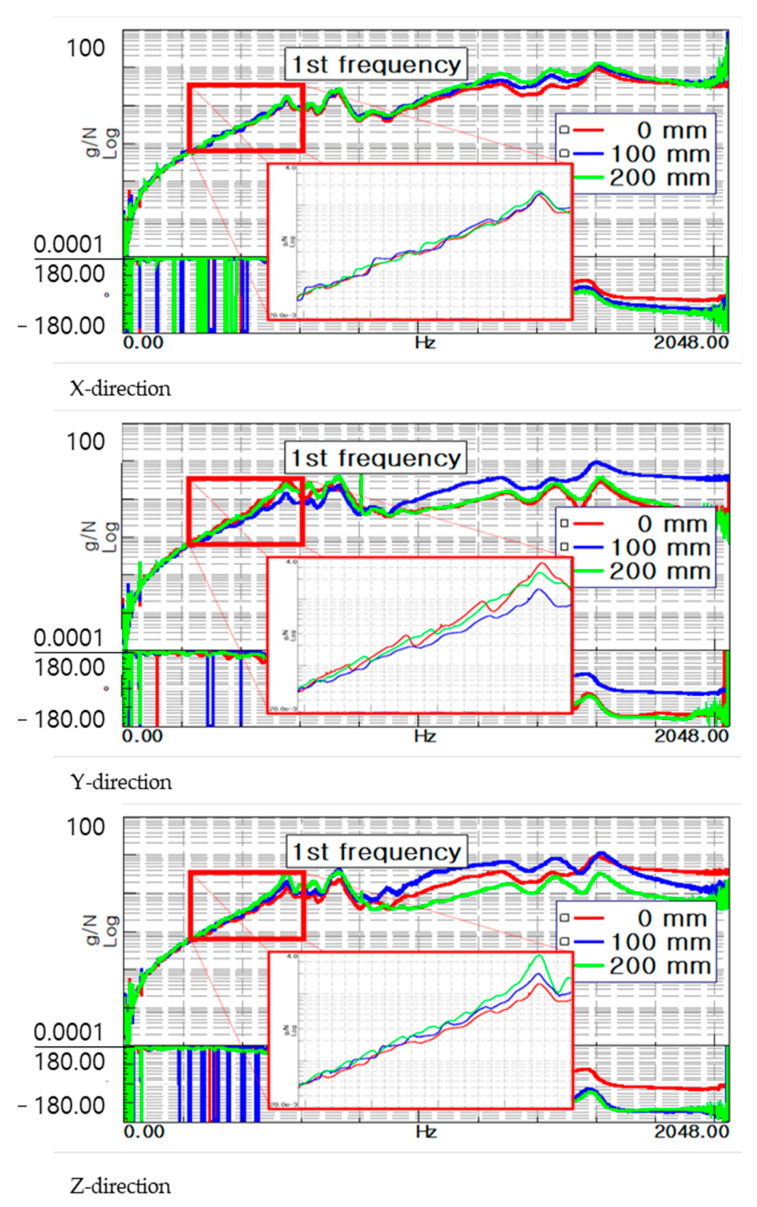
FRF based on direction and position.

**Figure 4 materials-15-07158-f004:**
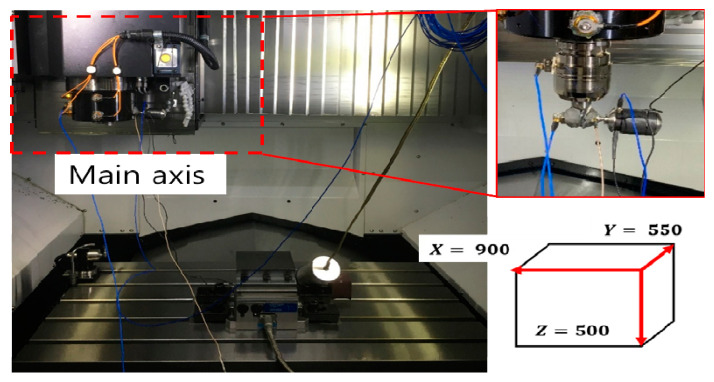
Experimental setup for design of experiments.

**Figure 5 materials-15-07158-f005:**
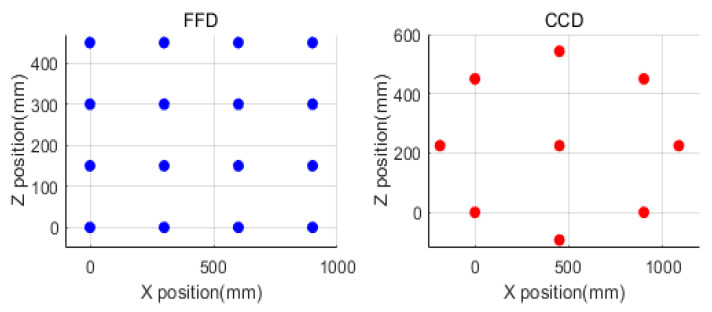
Experimental points based on design of experiments.

**Figure 6 materials-15-07158-f006:**
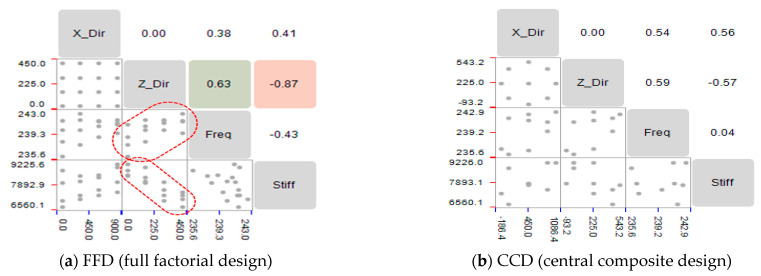
Comparison of correlation analyses.

**Figure 7 materials-15-07158-f007:**
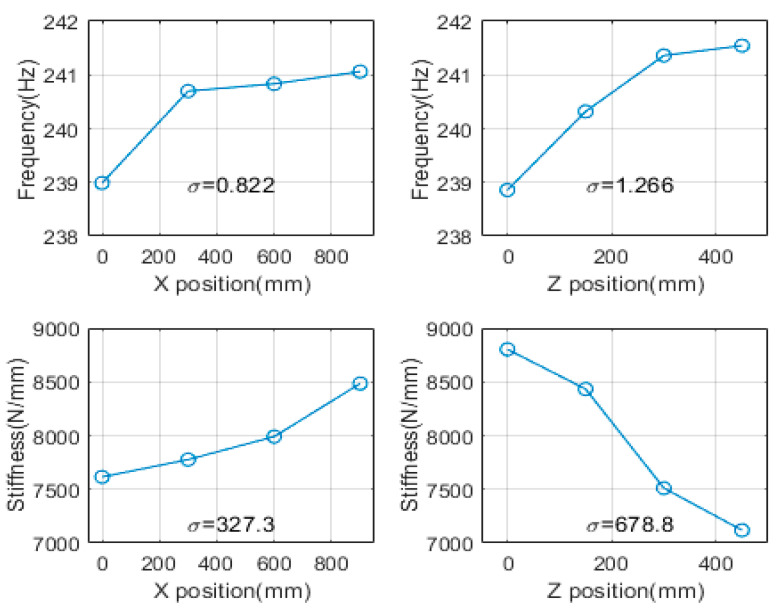
Comparison of ANOM based on spindle position (resonant frequency and dynamic stiffness).

**Figure 8 materials-15-07158-f008:**
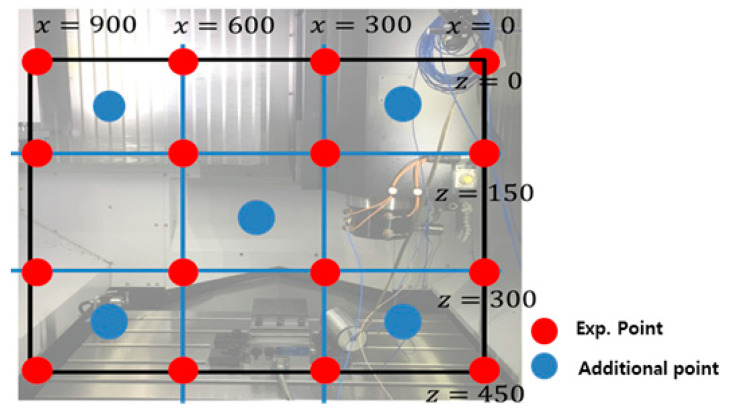
Experimental and additional points.

**Figure 9 materials-15-07158-f009:**
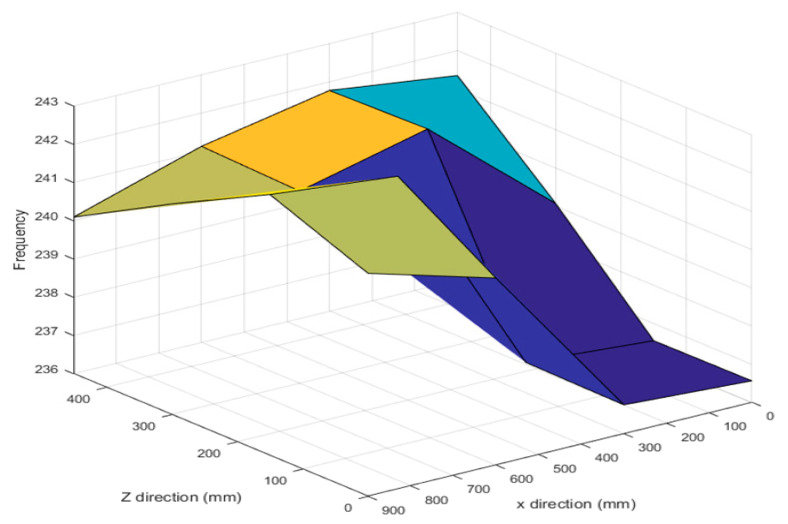
First resonant frequency distribution based on spindle position.

**Figure 10 materials-15-07158-f010:**
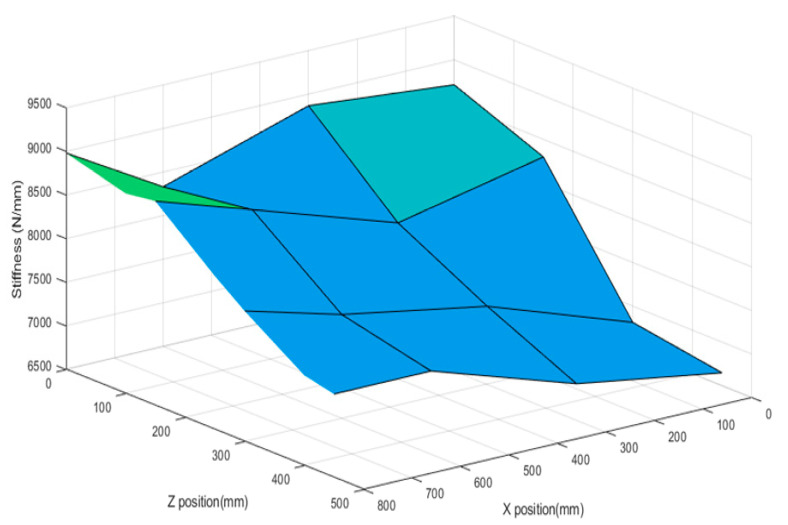
Dynamic stiffness distribution based on spindle position.

**Figure 11 materials-15-07158-f011:**
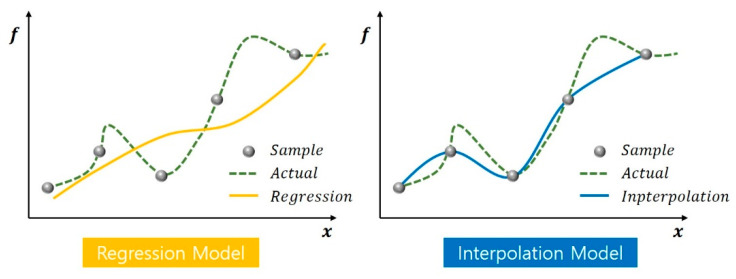
Comparison between regression and interpolation models [31].

**Figure 12 materials-15-07158-f012:**
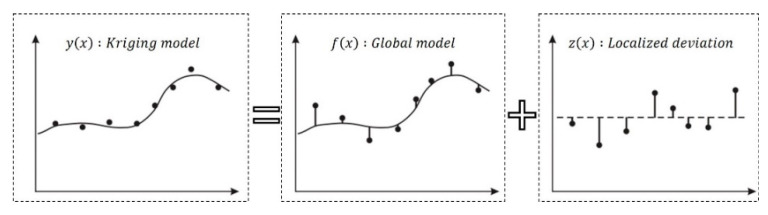
Kriging interpolation model [31].

**Figure 13 materials-15-07158-f013:**
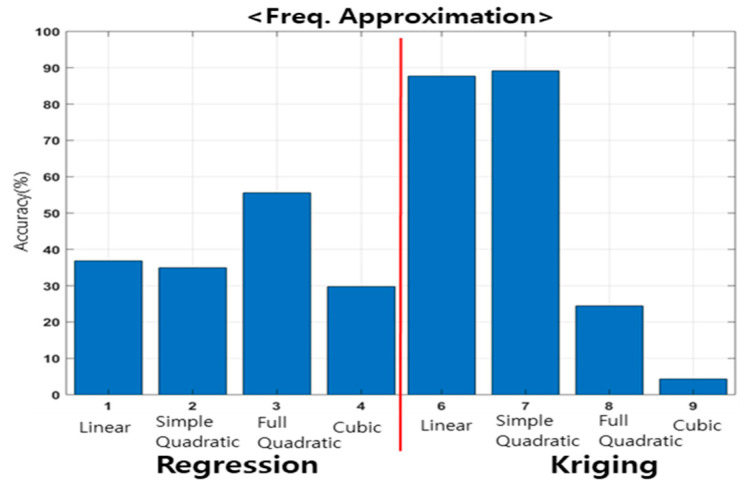
Comparison between approximation models for investigating resonant frequency.

**Figure 14 materials-15-07158-f014:**
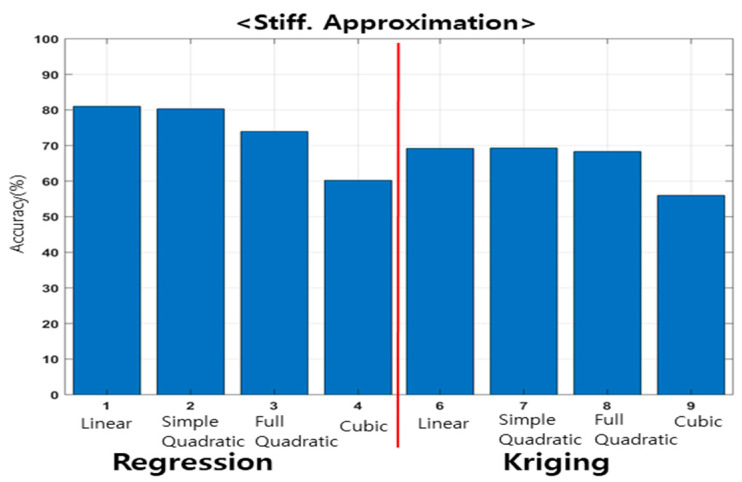
Comparison between approximation models for investigating dynamic stiffness.

**Figure 15 materials-15-07158-f015:**
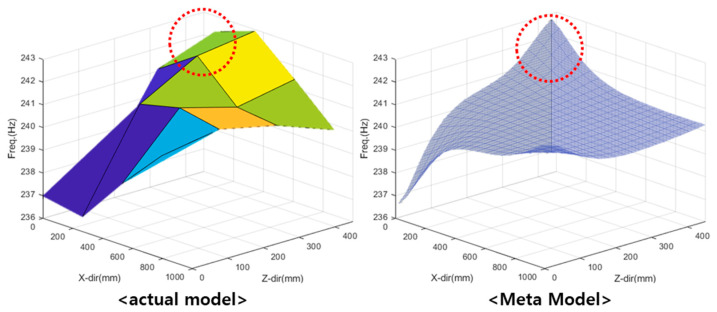
Comparison between actual and simple quadratic kriging models for investigating resonant frequencies.

**Figure 16 materials-15-07158-f016:**
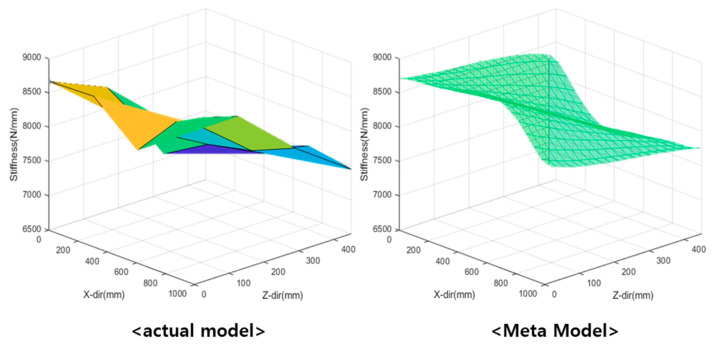
Comparison between actual and linear regression models for investigating dynamic stiffness.

**Table 1 materials-15-07158-t001:** Specifications of integral shakers [15].

Name	Qsource Integral Shaker
Frequency range (random test)	20–2000 Hz
Force level	7 Nrms
type	Integrated circuit piezoelectric

**Table 2 materials-15-07158-t002:** Experimental conditions.

Parameters	Value
Sampling rate (Hz)	4096
Input signal	Burst random
Window	Hanning
FRF estimation	Hv

**Table 3 materials-15-07158-t003:** Experimental condition for approximation model.

Parameter	Value
Sampling rate (Hz)	4096
Input signal	Burst random
Window	Hanning
FRF estimation	Hv

**Table 4 materials-15-07158-t004:** Validation of resonant frequency simple quadratic kriging model for additional experimental points.

Resonant Frequency (Hz)			
Position	Actual	Kriging	Error Rate
(150, 75)	236.688	237.476	0.3%
(150, 375)	242.730	241.694	0.4%
(450, 225)	236.466	240.283	1.6%
(750, 75)	238.373	240.262	0.7%
(751, 375)	241.338	240.775	0.2%

**Table 5 materials-15-07158-t005:** Validation of dynamic stiffness linear regression model for additional experimental points.

Dynamic Stiffness (N/mm)			
Position	Actual	Regression	Error Rate
(150, 75)	7624.00	8168.52	7.1%
(150, 375)	6794.05	7035.59	3.5%
(450, 225)	8076.28	7905.72	2.1%
(750, 75)	8442.35	8775.86	3.9%
(751, 375)	7624.02	7642.93	0.2%

## Data Availability

The data presented in this study are available on request from the corresponding author and the first author.

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
