# Peer review of "Approximation Model Development and Dynamic Characteristic Analysis Based on Spindle Position of Machining Center"

_materials, 2022, doi:10.3390/ma15207158_

Round 1

Reviewer 1 Report

This article studies the dynamic performance of the spindle at different positions. But I am afraid not enough innovations are shown in the article. In my opinion, what the author studies is a simple fitting or regression problem. The author needs to further highlight the innovation of the research methods. 

Author Response

We would like to thank you and the reviews of the MDPI for taking the time to review our article. We have made some corrections and clarifications in the manuscript after going over the reviews’ comments. 
There is some reason in what reviewer pointed out. The scope of this study was to develop an approximate model of the change in dynamic characteristics according to the position of the spindle. For dynamic characteristics, resonant frequency and dynamic stiffness were used. For high-precision and high-quality processing of machining processing, it is essential to analyze the dynamic characteristics of the machine. Also, the dynamic characteristics of the machine are required for stability lobe diagram analysis. Previous studies were conducted by analyzing only the dynamic characteristics according to the tool and process conditions. However, when the position of the main spindle changes, the shape of the structure changes, and accordingly, the mass distribution changes and the dynamic characteristics of the structure change. Previous studies do not consider changes in the dynamic characteristics of the machine itself. Therefore, in this study, the dynamic characteristics according to the change of the shape of the structure were analyzed. Vibration test and dynamic characteristics were analyzed for two machines, and the frequency and dynamic stiffness according to the change in the position of the main spindle were evaluated. An approximate model was developed using regression and interpolation to analyze the dynamic characteristics of frequency and stiffness at all positions of the main spindle. Interpolation used kriging among the most suitable approximate function evaluation methods for experimental data. We have added more detailed descriptions and mathematical formulas for kriging method and Related literature. Additional explanations have been added for the accuracy evaluation method and related expressions have been added.

Reviewer 2 Report

The following are my concerns. 

1. The quality of the pictures is not high, and some are blurred. Suggest the author enhance the quality of the pictures and optimize them.

2. Insufficient theoretical analysis. Suggest the author increase the theoretical analysis and derivation.

3. It is suggested that the authors add the data comparison with other methods.

4. It is suggested that the authors increase the references to the cutting-edge work of other researchers in recent years.

Author Response

We would like to thank you and the reviews of the MDPI for taking the time to review our article. We have made some corrections and clarifications in the manuscript after going over the reviews’ comments. The changes are summarized below: Responses to the review’s comments.

Comment 1 : The quality of the pictures is not high, and some are blurred. Suggest the author enhance the quality of the pictures and optimize them.

Our response : We have corrected the low-quality figure 1 and figure 2 into high-quality picture as the reviewer suggested.

Coment 2 :
- Insufficient theoretical analysis. Suggest the author increase the theoretical analysis and derivation.
- The paper does not introduce enough theory and analysis of the model. The authors can add pictures of the theoretical description. Also, the description of the mathematical model needs to be added. The paper currently only provides a summation formula. This is of minimal help to the description of the model. The authors should add the mathematical derivation related to model building, result testing, and reliability analysis.

Our response : 
We agree with the reviewer about that. We have added theoretical derivation for compliance in chapter 2 and kriging method in chapter 4.2, estimation of accuracy in chapter 4.3.
The changes are summarized in the Word file: Responses to the review’s comments.

Comment 3 : 
- It is suggested that the authors add the data comparison with other methods.
- The experimental results of the paper do not compare the data with other methods, and the credibility of the model effect is not high. The authors should add data comparisons with other existing methods in the experimental results section and provide the necessary graphs to demonstrate the model effects.

Our response : 
We agree with the reviewer about that. However, the scope of this study was an approximate model of resonant frequency and dynamic stiffness, comparing regression and interpolation. For regression models, linear polynomial, simple quadratic polynomial, full quadratic polynomial, simple cubic polynomial, and interpolation models were also compared with linear kriging, simple quadratic kriging, full quadratic kriging, and simple cubic kriging. 
We compare the results of regression and kriging methods for resonant frequency and dynamic stiffness, and for resonant frequency, simple quadratic kriging shows an accuracy of about 89%. As for the dynamic stiffness, linear regression shows an accuracy of about 81%. Because the accuracy was evaluated with the r-square method, the overall result shows a rather low accuracy. However, when comparing each point through additional experimental points, the resonant frequency model has an error of about 1% and the dynamic stiffness shows an error of less than 7%. The accuracy of each point is more than 90%, which shows that it is suitable for fitting experimental data.

Comment 4 : 
It is suggested that the authors increase the references to the cutting-edge work of other researchers in recent years.

Our response :
We have added the references to the machining cutting and dynamic characteristic of machining in to introduction as the reviewer suggested.
The changes are summarized in the Word file: Responses to the review’s comments.

Reviewer 3 Report

The article is devoted to the issues of planning the experiment, its execution and the analysis of the results. The issue concerns dynamic characteristic analysis based on spindle position of machining center.

Generally correct article, although only indirectly related to Materials.

Remarks to consider:

1. The concept of "accuracy" should be defined more precisely. The authors do not examine the "accuracy model" starting from the internal structure of the model and its numerous parameters, but only carry out an undefined comparative analysis.

2. The terms "the meta model" and "the kriging model" should be clarified. Is it the same or is it different?

3. The conclusions are in fact a summary. There are no bulleted conclusions from the research.

Author Response

We would like to thank you and the reviews of the MDPI for taking the time to review our article. We have made some corrections and clarifications in the manuscript after going over the reviews’ comments. The changes are summarized below: Responses to the review’s comments.

Comment 1 : The concept of "accuracy" should be defined more precisely. The authors do not examine the "accuracy model" starting from the internal structure of the model and its numerous parameters, but only carry out an undefined comparative analysis.

Our response : We agree with the reviewer about that. We have corrected the “accuracy”. In addition, we have added examine the accuracy in chapter 4.3. We have used r-square for accuracy. The closer the r-square is to 1, the higher the accuracy, and in this study, it was expressed as a percentage. We have denoted r-square in Equation 14(page .11)
The changes are summarized in the Word file: Responses to the review’s comments.

Comment 2 : The terms "the meta model" and "the kriging model" should be clarified. Is it the same or is it different?

Our response : Meta model is different from kriging model. Meta model and approximate model were mixed and used. We have modified the term meta model in the paper to approximate model. The kriging model is an interpolation model. As shown in Section 4.2, approximate models are largely divided into regression and interpolation. The Kriging method is a representative model of the interpolation model.  We have added mathematical formulas of kriging methods and related references to Section 4.2.

Comment 3 : The conclusions are in fact a summary. There are no bulleted conclusions from the research.

Our response : We agree with the reviewer about that. We have corrected conclusions. The changes are summarized in the Word file: Responses to the review’s comments.

Reviewer 4 Report

The authors proposed the development of a Meta model for predicting the dynamic characteristics of a milling tool at different positions in working space.

The metal model was developed by kriging method based on experimental data collected from milling center and has been verified with high accuracy in predicting the dynamic characteristics of the milling tool at different positions. Overall, there are some deficiencies needed to address in details, as below:
(1) As shown in Figure 2, a
ccelerometers were mounted on tool tip, did authors consider the mass effect of sensors on the vibration amplitude, which may thus affect the evaluation of the dynamic characteristics to be examined.

(2) There is no clear justification on the frequency ranges and modal parameters taken as the basis for model development.  

  It is noted from measured FRFs, the frequency ranging from 236 to 242 Hz did not show significant vibration amplitude, which cannot sufficiently reflect the dynamic characteristics of the machine tool that have substantial influences on machining performances.

 (3) Please provide the mathematical formulas of the prediction models established by regression, interpolation and kriging method. It would be helpful to examine the superiority of the Metal model proposed in this study.

(4) Again, regarding the dynamic characteristics of the milling tool, I am wondering why tool point FRFs were not assessed by tapping test (or impact test), which are well recognized an effective way to characterize the dynamic characteristics of milling tool, including the natural frequency, modal damping and dynamic stiffness at different vibration modes and frequency ranges. Also these modal parameters are more useful in identify the machining performance at different frequency ranges. Authors are encouraged to consider this point and present new findings.

(5)Section "Introduction" can be enhanced by including more literatures relating to the assessment or prediction of the dynamic characteristics with its influences on machining performances of machine tool.  

(6)On Page 1, line 39, Citations “1-5” are included in a sentence; they should be explained with specific and correct details. Similarly, on page 2, line 57 citations "6-9".

Author Response

We would like to thank you and the reviews of the MDPI for taking the time to review our article. We have made some corrections and clarifications in the manuscript after going over the reviews’ comments. The changes are summarized below: Responses to the review’s comments.

Comment 1 : As shown in Figure 2, accelerometers were mounted on tool tip, did authors consider the mass effect of sensors on the vibration amplitude, which may thus affect the evaluation of the dynamic characteristics to be examined.

Our response : As the reviewer mentioned, the Accelerometer and shaker have mounted to tool tip. We agree with the reviewer about that. However, we have evaluated the dynamic characteristics of the entire machining center, not the vibration or dynamic characteristics of the tool. The weight of the Accelerometer and Shaker is about 350g, which is very small compared to the entire machining center, so we have thought the mass effect can be neglected. As mentioned in Ref.12, more accurate stiffness evaluation is possible when the excitation test is used rather than the impact test. 

Comment 2 : There is no clear justification on the frequency ranges and modal parameters taken as the basis for model development. 
It is noted from measured FRFs, the frequency ranging from 236 to 242 Hz did not show significant vibration amplitude, which cannot sufficiently reflect the dynamic characteristics of the machine tool that have substantial influences on machining performances.

Our response : We agree with the reviewer about that. However, the 1st resonant frequency of the machining tool occurs at about 2kHz or higher, and in the case of small machining center, the resonant frequency occurs between about 200-500Hz. This is stated in references 12 and 13. Therefore, we have selected the frequency band below 2kHz and set the first natural frequency to the band of 200Hz. We have thought the vibration amplitude is small because it is the response of the whole machine tool system.
Although there is not much difference in resonant frequency, when analyzing the stability lobe diagram, even a small difference in frequency significantly affects the stability of cutting depth and cutting speed in case of very precise machining. Even in the case of dynamic stiffness, a difference of up to about 2500N/mm occurs, which greatly affects machinability. The modal parameter, which is the dynamic stiffness that identifies the dynamic characteristics, was additionally written in Chapter 2.

Comment 3 : Please provide the mathematical formulas of the prediction models established by regression, interpolation and kriging method. It would be helpful to examine the superiority of the Metal model proposed in this study.

Our response : We agree with the reviewer about that. We have added theoretical derivation for kriging method in chapter 4.2.
The changes are summarized in the Word file: Responses to the review’s comments.

Comment 4 : Again, regarding the dynamic characteristics of the milling tool, I am wondering why tool point FRFs were not assessed by tapping test (or impact test), which are well recognized an effective way to characterize the dynamic characteristics of milling tool, including the natural frequency, modal damping and dynamic stiffness at different vibration modes and frequency ranges. Also these modal parameters are more useful in identify the machining performance at different frequency ranges. Authors are encouraged to consider this point and present new findings.

Our response : According to the reviewer's opinion, the impact test is an effective method to identify the dynamic characteristics of the tools of a machine, but the excitation test is effective when focusing on the dynamic characteristics of the machine. In addition, Simcenter's qsource has an accelerometer attached to the exciter, so it is suitable for driving point FRF. 
In the case of the impact test, there is a spatial difficulty in attaching an accelerometer to the tip of the tool to make an impact. vibrating force is applied to the tip of the tool. If a general tool is used without developing a dummy tool, there is a possibility that the force may be transmitted in a different direction due to the angle of the tool blade.
The impact test does not have enough force to transmit the excitation force to the entire machine. However, in the case of an exciter, sufficient excitation force can be delivered by using an amplifier.
For DOE, several repeated experiments are required, and frequency response functions are derived through multiple excitations and responses per point. Since reproducibility is essential for a uniform experiment, the excitation test was performed.

Comment 5 : Section "Introduction" can be enhanced by including more literatures relating to the assessment or prediction of the dynamic characteristics with its influences on machining performances of machine tool.

Our response : We agree with the reviewer about that. We have added relevant literature.
The changes are summarized in the Word file: Responses to the review’s comments.

Comment 6 : On Page 1, line 39, Citations “1-5” are included in a sentence; they should be explained with specific and correct details. Similarly, on page 2, line 57 citations "6-9".

Our response : We agree with the reviewer about that. we have added a specific and details citation in chapter 1 and 2.
The changes are summarized in the word file: Responses to the review’s comments. 

Round 2

Reviewer 1 Report

Thanks for the author's explanation, I agree that this article can be accepted now

Author Response

We would like the thank you and the reviewer of the MDPI for taking the time to review our article.
We are proofread in newly add text in article.

Reviewer 3 Report

The article presents in a concise and understandable way the model development and dynamic characteristic analysis based on spindle position of machining center. I propose to accept the article as it is.

Author Response

(The authors gave the same response as above.)

Reviewer 4 Report

I am pleased to acknowledge that some of major concerns about the manuscript content have been appropriately addressed with more descriptions. In my opinion, the article is publishable, but final proofreading in newly added text is needed.

Author Response

(The authors gave the same response as above.)
